# Determining Response to Treatment for Drug-Induced Bronchocentric Granulomatosis by the Forced Oscillation Technique

**DOI:** 10.3390/medicina57121315

**Published:** 2021-11-30

**Authors:** Susumu Fukahori, Yasushi Obase, Chizu Fukushima, Daisuke Takao, Jun Iriki, Mutsumi Ozasa, Yoshiaki Zaizen, Noboru Takamura, Junya Fukuoka, Kazuto Ashizawa, Hiroshi Mukae

**Affiliations:** 1Department of Respiratory Medicine, Nagasaki University Hospital, 1-7-1 Sakamoto, Nagasaki 852-8501, Japan; susumu-f@nagasaki-u.ac.jp (S.F.); f-chizu@nagasaki-u.ac.jp (C.F.); dtakao565@gmail.com (D.T.); jiriki123456@gmail.com (J.I.); hmukae@nagasaki-u.ac.jp (H.M.); 2Department of Respiratory Medicine, Graduate School of Biomedical Sciences, Nagasaki University, 1-7-1 Sakamoto, Nagasaki 852-8501, Japan; 3Department of Pathology, Nagasaki University Hospital, 1-7-1 Sakamoto, Nagasaki 852-8501, Japan; ozasamu@nagasaki-u.ac.jp (M.O.); zaizen_yoshiaki@med.kurume-u.ac.jp (Y.Z.); fukuokaj@nagasaki-u.ac (J.F.); 4Atomic Bomb Disease Institute, Nagasaki University, 1-12-4 Sakamoto, Nagasaki 852-8523, Japan; takamura@nagasaki-u.ac.jp; 5Department of Clinical Oncology, Graduate School of Biomedical Sciences, Nagasaki University 1-7-1 Sakamoto, Nagasaki 852-8501, Japan; ashi@nagasaki-u.ac.jp

**Keywords:** forced oscillation technique, small airway disease, anti-tumor necrosis factor alpha therapy, sarcoid-like reaction, bronchocentric granulomatosis, ulcerative colitis, inflammatory bowel disease, golimumab

## Abstract

Anti-tumor necrosis factor alpha (TNFα) therapy is widely used to treat various inflammatory conditions. Paradoxically, there are several case reports describing the development of bronchocentric granulomatosis treated with TNFα inhibitors, and it is difficult to determine the effect of treatment using conventional spirometry because the lesions are located in small airways. However, it has been reported that the forced oscillation technique (FOT) is useful in the evaluation of small airway disease in bronchial asthma or chronic obstructive pulmonary disease. We performed the FOT to determine the effect of treatment on bronchocentric granulomatosis and found it to be useful. We report the case of a 55-year-old female with ulcerative colitis who was treated with golimumab and who developed bronchocentric granulomatosis as a sarcoid-like reaction to golimumab. She was successfully treated with prednisone, and the treatment efficacy was confirmed by the FOT. The FOT may be useful in the evaluation of small airway disease in bronchocentric granulomatosis. This case may help inform clinicians of the usefulness of the FOT to assess small airway disease in various diseases.

## 1. Introduction

Several types of anti-tumor necrosis factor alpha (TNFα) therapy are widely used for inflammatory bowel disease (IBD) [1] and have demonstrated variable therapeutic efficacy in inflammatory conditions such as rheumatoid arthritis [2], psoriatic arthritis [3], and sarcoidosis [4]. Several side effects of anti-TNFα therapy, including various infections and inflammatory responses, have been reported [5]. Golimumab is a human monoclonal antibody that acts as an anti-inflammatory drug by inhibiting TNFα, which can induce inflammatory cell lysis by activating complement and antibody-mediated cytotoxicity and promoting T-cell response, consequently inhibiting the inflammatory cascade [6]. Although TNFα antagonists are effective in treating inflammatory conditions such as sarcoidosis, a paradoxical sarcoid-like reaction has been observed in approximately 1/2800 patients treated for inflammatory arthropathies [7].

Herein, we report the case of a 55-year-old female with ulcerative colitis (UC) who was treated with golimumab and developed bronchocentric granulomatosis as a sarcoid-like reaction to golimumab. She was treated with prednisone, and her dyspnea was reduced. However, in this case, the lesions were mainly distributed in the small airways (internal diameter <2 mm), making it difficult to objectively evaluate the efficacy of conventional spirometry treatment [8]. Since the forced oscillation technique (FOT) has been reported to be useful in the evaluation of small airway disease [9], we used the FOT to determine the response to treatment in this case. Here, we report the usefulness of the FOT in determining the improvement of the small airway disease in bronchocentric granulomatosis.

## 2. Case Presentation

A 55-year-old female with a 2-year history of UC was referred to our hospital for further assessment of dyspnea on exercise (modified Medical Research Council (mMRC) grade 4) and non-productive cough. She had no history of asthma or dust exposure and had never smoked tobacco.

Her initial treatment for UC was mesalazine monotherapy 20 months prior to her referral. Due to diarrhea and vomiting most likely caused by mesalazine, mesalazine was switched to salazopyrin 18 months prior. Subsequently, salazopyrin was switched to golimumab due to liver dysfunction and eosinophilia, most likely caused by salazopyrin 12 months prior. Ten months after golimumab was introduced, she developed dyspnea on exercise and a non-productive cough. Physical examination revealed tachypnea and wheezing in both lungs on auscultation. Chest radiography and spirometry were performed. Chest radiography was normal, and an obstruction was found on spirometry.

She had been diagnosed with bronchial asthma, and treatment with inhaled corticosteroids and long-acting beta agonists were administered for 2 months. However, the respiratory symptoms did not improve, and so the referring physicians considered if these respiratory symptoms were caused by golimumab. Golimumab was switched to vedolizumab, and she was referred to our hospital for further examination.

Physical examination on admission revealed a temperature of 36.0 °C, blood pressure of 122/78 mmHg, and a regular pulse of 72 bpm. Lung auscultation revealed normal vesicular sounds in both the lungs. Laboratory findings included the following: white blood cells, 5900/μL; hemoglobin, 14.9 g/dL; lactate dehydrogenase, 405 IU/L; Krebs von den Lungen-6 level, 2154 U/mL. Biological findings showed a normal hematological panel and absence of inflammatory activation. Liver and renal function test results were normal. The markers for HTLV-1, hepatitis B virus, and hepatitis C virus infections were negative. Angiotensin converting enzyme, rheumatoid factor, antinuclear antibodies, and myeloperoxidase anti-neutrophil antibody test results were within normal ranges. Arterial blood gas analysis under ambient air showed a pH of 7.419, HCO_3_ of 23.9 mmol/L, PaCO_2_ of 37.6 mmHg, and PaO_2_ of 73.3 mmHg.

An acid-fast bacilli (AFB) smear and culture tests and *Mycobacterium tuberculosis* polymerase chain reaction (PCR) of a sputum specimen test were negative. The T-SPOT interferon γ-release assay was negative.

High-resolution computed tomography (on first visit) showed mosaic attenuation, which might reflect decreased perfusion of the poorly ventilated regions and multiple small centrilobular and branching nodules in both lungs (Figure 1).

The findings on spirometry were a decreased forced vital capacity (FVC: 1.18 L (44% of predicted)), a reduced forced expiratory volume in 1 s (FEV_1_: 0.52 L (22% of predicted)), and a reduced ratio of FEV_1_ to FVC (44.1%), with a poor response to inhaled bronchodilators (post-bronchodilator FEV_1_: 0.53 L).

Unfortunately, the diffusing capacity for carbon monoxide (D_L_CO) data were not obtained because of her low pulmonary function.

Respiratory system resistance (Rrs) and respiratory system reactance (Xrs) were measured with a broadband FOT using a commercially available device (MostGraph-01; Chest M.I. Co. Ltd., Tokyo, Japan). The mouth pressure and flow signals were measured and calculated to obtain the Rrs and Xrs properties against oscillatory frequencies ranging from 4 to 36 Hz. Rrs during exhalation was 0.56 kPa/L/s (Ex) at 5 Hz and 0.45 kPa/L/s (Ex) at 20 Hz, mean R5-R20 was 0.11 kPa/L/s (the presence of small airway disease is suggested by (R5-R20) > 0.07 kPa/L/s; [9]). Xrs during exhalation was −0.38 kPa/L/s (Ex) at 5 Hz. ALX (low-frequency reactance area) during exhalation was 27.8 kPa/L.

Bronchoalveolar lavage fluid (BALF) was obtained from the right middle lobe (B^5b^). AFB smear and culture and *M. tuberculosis* PCR of BALF tests were negative. BALF cultures and stains were negative for other infectious organisms. An analysis of BALF revealed an increased percentage of lymphocytes (67%) and a CD4/CD8 ratio of 5.2.

After BAL, a transbronchial lung biopsy of the left upper lobe of the lung was performed; however, the histological pattern was not specific. There was no evidence of non-caseating epithelioid granulomas. Moreover, there was no evidence of eosinophil infiltration.

Radiological findings and transbronchial biopsy findings did not allow for diagnosis. Hence, a surgical open lung biopsy was performed. Surgical open lung biopsy specimens revealed non-necrotizing granulomas, suggesting bronchocentric granulomatosis. Histopathological stains were negative for malignancy, and no AFB or fungal elements were identified (Figure 2).

The bronchocentric granulomatosis seen in this case could have been caused by pulmonary involvement of UC. However, when the treatment of UC was changed from golimumab to vedolizumab, the UC itself improved significantly, but the bronchocentric granulomatosis remained, so we diagnosed this as a side effect of golimumab. Initial treatment with prednisone was then introduced at a dose of 30 mg per day (0.5 mg/kg/day).

After four days of treatment, her dyspnea was reduced (mMRC grade 4 to grade 2), but she developed steroid psychosis. As she could no longer undergo proper spirometry testing, we decided to use an FOT to check her response to corticosteroid therapy. After one month of therapy, respiratory resistance measurements (R5-R20, X5, ALX) based on the FOT improved (Figure 3). The response to corticosteroid therapy was evidenced by improvement of dyspnea and a decrease in airway resistance at 1 month.

The prednisone dose was then titrated. The patient continued oral corticosteroid therapy at a dose of 5 mg of prednisone per day.

## 3. Discussion

Development of sarcoidosis-like granulomatosis in patients treated with TNFα antagonists is a phenomenon previously under-recognized. Since the introduction of anti-TNFα therapy in several diseases such as rheumatoid arthritis, psoriasis, and IBD, this reaction has been increasingly reported. This presentation is paradoxical, since anti-TNFα therapy can also be used in refractory cases of sarcoidosis [10,11]. To date, the development of sarcoidosis-like lesions under anti-TNFα is a class effect rather than being drug-specific [12]. Although the precise mechanism of the sarcoidosis-like reaction caused by anti-TNFα therapy is largely unknown, there are several hypotheses to explain this paradoxical reaction.

Cytokine imbalance due to long-term TNFα suppression [13] and the role of an underlying infectious agent [14] have been suggested.

In this case, dyspnea diminished after corticosteroid therapy was started, and the efficacy of the treatment was confirmed by the FOT.

Pulmonary granulomas can present primarily in the small airways in cases with sarcoidosis and sarcoid-like reactions; therefore, sometimes it may be difficult to detect the response to therapy for these diseases by traditional pulmonary function tests using spirometry. Peak expiratory flow and FEV_1_ mainly reflect the function of large airways; however, FEV_25–75_ is believed to represent the function of small airways [15], but it suffers from poor reproducibility [16].

On the other hand, the FOT requires minimal patient cooperation and can be done easily in subjects who are unable to perform spirometry. In addition, there is evidence that the FOT is more sensitive than spirometry for detecting bronchodilator effects in patients with asthma and chronic obstructive pulmonary disease [17,18]. Recently, it was reported that the FOT parameters correlated better with clinical symptoms and asthma control than spirometry indices in asthmatic patients [19]. Furthermore, it has been reported that the FOT is more sensitive than spirometry in identifying pathologies in the peripheral airways [9,20]. The main limitation of the FOT is that minor changes in glottal aperture and small buccal airflow leaks can significantly affect the FOT [21]. In this case, the FOT was beneficial for detecting the therapeutic effects of the therapy.

## 4. Conclusions

In conclusion, if a patient being treated with TNFα inhibitors presents with respiratory symptoms such as wheezing, dyspnea, or dry cough and no abnormalities can be detected by simple chest imaging or spirometry, bronchocentric granulomatosis must be considered as a differential diagnosis. The FOT should be performed to help diagnose the presence of bronchocentric granulomatosis in small airways and to help determine the response to treatment.

## Figures and Tables

**Figure 1 medicina-57-01315-f001:**
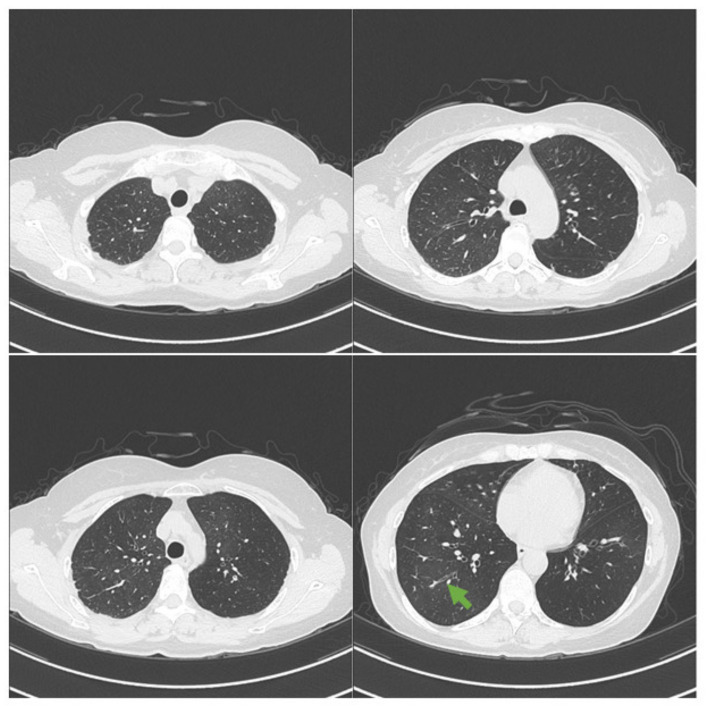
Computed tomography shows mosaic attenuation and multiple small centrilobular and branching nodules in both lungs. Bronchial wall thickening and lumen narrowing at the peripheral side were also seen (green arrows).

**Figure 2 medicina-57-01315-f002:**
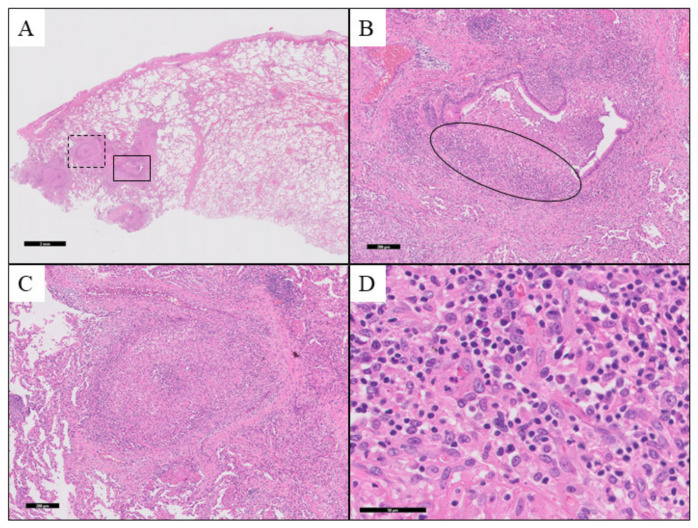
Pathological findings of the surgical lung biopsy. (**A**) Membranous to respiratory bronchioles (hematoxylin and eosin (H&E) stain; magnification, ×5). (**B**) The lower right enclosed area of Panel (**A**). The airway becomes erosive, and histiocytes enter the airway and form granules. Inflammatory cell infiltration, mainly by lymphocytes and plasma cells, was observed around the erosions (H&E stain; magnification, ×50). (**C**) The upper left enclosed area of Panel (**A**). Granulomas are seen around the ruptured bronchus. There is little necrosis inside, but neutrophil infiltrates. (**D**) Enclosed area in (**B**). The inflammatory cells that infiltrate the airway near the erosions are lymphocytes and plasma cells.

**Figure 3 medicina-57-01315-f003:**
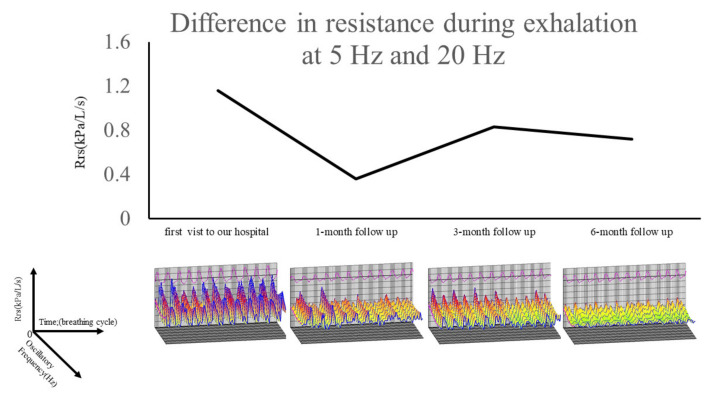
Changes of R5-R20 (Ex) and colored three-dimensional images of respiratory cycle dependence (Rrs was higher and Xrs was more negative in the expiratory phases than in the inspiratory phases) and frequency dependence (Rrs increased at lower frequencies and fell with increasing frequencies) over a 6-month period. R5-R20 (Ex), difference in resistance during exhalation at 5 Hz and 20 Hz; Rrs, respiratory system resistance; Xrs, respiratory system reactance.

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
