# Peer review of "Determining Response to Treatment for Drug-Induced Bronchocentric Granulomatosis by the Forced Oscillation Technique"

_medicina, 2021, doi:10.3390/medicina57121315_

Round 1

Reviewer 1 Report

I have read the case report by Fukahori et al. with great interest. The authors used FOT when following on the course of bronchocentric granulomatosis.

Comments:

  • Lines 19 and 50. FOT is not new technique (developed in 1950s). Please, delete “recently”.
  • You should emphasise the advantage of FOT over lung function test, such as it can be performed during general anaesthesia, and in people who cannot perform spirometry reproducibly.
  • Apart from R5-R20, FOT records other parameters. Please report X5, AX and their changes following the treatment.
  • As mentioned, FOT is a very sensitive technique. Upper airway artefacts, such as leakage if the mouth is not properly sealed or glottal closure could it affect the values (https://pubmed.ncbi.nlm.nih.gov/25742459/). Please discuss this limitation and acknowledge the paper.
  • Line 170. FOT not necessarily more sensitive than spirometry when correlating with symptoms. Some patients with symptoms and spirometric changes do not have significant changes in FOT values (https://pubmed.ncbi.nlm.nih.gov/26291140/). Please amend the sentence, appropriately.

Author Response

Reviewer 1

Comment

Lines 19 and 50. FOT is not new technique (developed in 1950s). Please, delete “recently”.

Response

Thank you for your comment. As suggested, we have revised the sentence as follows:

 " However, it has recently been reported that the forced oscillation technique (FOT) is useful in the evaluation of small airway disease in bronchial asthma or chronic obstructive pulmonary disease. We performed FOT to determine the effect of treatment on bronchocentric granulomatosis and found it to be useful." Page 1 lines 19-22

"Since the forced oscillation technique (FOT) has recently been reported to be useful in the evaluation of small airway disease [9], we used FOT to determine the response to treatment in this case. " Page 2 lines 49-51

Comment

You should emphasise the advantage of FOT over lung function test, such as it can be performed during general anaesthesia, and in people who cannot perform spirometry reproducibly.

Response

Thank you for your comment. As suggested, we have added this sentence to read as follows: " On the other hand, FOT requires minimal patient cooperation and can be done easily in subjects who are unable to perform spirometry. 

Also, there is evidence that FOT is more sensitive than spirometry for detecting bronchodilator effects in patients with asthma and chronic obstructive pulmonary disease "

Page 5 lines 179-182

Comment

Apart from R5-R20, FOT records other parameters. Please report X5, AX and their changes following the treatment.

Response

Thank you for your comment. As suggested, we have added information about X5 and ALX.

Page 3 lines 107-108

Page 4, line 142

Comment

As mentioned, FOT is a very sensitive technique. Upper airway artefacts, such as leakage if the mouth is not properly sealed or glottal closure could it affect the values (https://pubmed.ncbi.nlm.nih.gov/25742459/). Please discuss this limitation and acknowledge the paper.

Response

Thank you for your comment. We have added the sentence as follows “The main limitation of FOT is that minor changes in glottal aperture and small buccal airflow leaks can significantly affect FOT.”

Page 5, Lines 182-183

Page 6, Line 184

Comment

Line 170. FOT not necessarily more sensitive than spirometry when correlating with symptoms. Some patients with symptoms and spirometric changes do not have significant changes in FOT values (https://pubmed.ncbi.nlm.nih.gov/26291140/). Please amend the sentence, appropriately.

Response

Thank you for your comment. As suggested, we have revised this sentence as follows: “Pulmonary granulomas can present primarily in the small airways in cases with sarcoidosis and sarcoid-like reactions; therefore, sometimes it may be difficult to detect the response to therapy for these diseases by traditional pulmonary function tests using spirometry.”

Page 5, lines 170-173

Reviewer 2 Report

In my opinion, prednizon not the forced oscillation technique was helpful in this patient.

Author Response

Reviewer 2

Comment

In my opinion, prednizon not the forced oscillation technique was helpful in this patient.

Response

Thank you for your comment. As you pointed out, it is certain that corticosteroid therapy not the FOT was helpful to reduce her dyspnea in this patient. But as reviewer 4 commented, we do believe that FOT was also helpful to monitor the patient's response to treatment.

Reviewer 3 Report

Dear authors,

it was the pleasure to read your paper. You have suggested to clinicians to perform FOT in patients with bronchial pathological processes. I will suggest you 3 minor suggestions:

  1. page 2 - add patient's "Respiratory rate" on addmision (in text after „regular pulse of 72/min“)
  2. write the comment on control CT chest when the FOT was performed (even it was not performed)
  3. write how long did the patient use 5 mg prednisone per day (in sentence „The patient continued oral corticosteroid therapy at a dose of 5 mg of prednisone per day.“)

Kind regards, reviewer

Author Response

Reviewer 3

Comment

page 2 - add patient's "Respiratory rate" on admission (in text after „regular pulse of 72/min“)

Response

Thank you for your comment. As suggested, we have added information about respiratory rate on admission.

Page 2 line 72

Comment

write the comment on control CT chest when the FOT was performed (even it was not performed)

Response

Thank you for your comment. Unfortunately, we did not perform a chest CT on the date the FOT was performed; on page 2, lines 85-87, you will find comments regarding chest CT findings performed on the first visit close to the date the FOT was performed.

Comment

write how long did the patient use 5 mg prednisone per day (in sentence „The patient continued oral corticosteroid therapy at a dose of 5 mg of prednisone per day.“)

Response

Thank you for your comment. As suggested, we have added the information about the duration for this patient continued corticosteroid therapy at a dose of 5 mg per day.

Page 5, Lines 155

Reviewer 4 Report

No additional comments. Nice case. Worth of publishing.

Author Response

Reviewer 4

Comment

No additional comments. Nice case. Worth of publishing.

Response

Thank you for your comment. We were very pleased to receive your positive evaluation of our manuscript.

Reviewer 5 Report

The paper present a case of a patient with ulcerative collitis, treated with TNF alfa inhibitors who developed sarcoidosis-like pulmonary granulomatosis. Due to severe respiratory disfunction, FOT was used to monitor the patient's response to treatment. The paper is very interesting and well written.

Minor issue :

-more information regarding FOT technique, results and limits in monitoring respiratory function should be provided în the discussion section 

Author Response

Reviewer 5

Comment

The paper presents a case of a patient with ulcerative colitis, treated with TNF alfa inhibitors who developed sarcoidosis-like pulmonary granulomatosis. Due to severe respiratory disfunction, FOT was used to monitor the paper is very interesting and well written. the patient's response to treatment.

Minor issue:

-more information regarding FOT technique, results and limits in monitoring respiratory function should be provided în the discussion section

Response

Thank you for your comment. As suggested, we have added the sentence as follows “The main limitation of FOT is that minor changes in glottal aperture and small buccal airflow leaks can significantly affect FOT.”

Page 5, Lines 182-183

Page 6, Line 184